# It's a Doge's Life: Examining Term Limits in Venetian Doges' Life Tenure

**Juan J. Merelo** 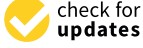

Department of Computer Engineering, Automation and Robotics, University of Granada, 18071 Granada, Spain; jmerelo@ugr.es

**Abstract:** During most of the lifespan of the Venetian republic, doges (the name their presidents received) were elected for life. However, a long tenure was a rare event, which effectively resulted in term limits, as has already been reported by several authors. In this paper, we examine the length of these tenures and their evolution during the existence of the Venetian republic, following Smith et al.'s claim that specific events in Venetian history caused this shortening, but also the dates and possibly event or events that effectively caused that limitation by design. Finally, we will discuss the causes of this limitation and its effective consequences.

**Keywords:** Venetian republic; office terms; computational history; Mediterranean history

## 1. Introduction

The Venetian republic, which existed from the seventh to the end of the eighteenth century, was a unique polity in many different aspects, including the fact that it lasted for more than one thousand years essentially under the same set of laws Martin and Romano (2003); its stability in a geographic milieu—the Italian peninsula—that underwent many waves of upheaval under the opposite forces of the Papal states; the Holy Roman Empire; and later the Ottoman, French, Spanish and Austro-Hungarian empires, Madden (2012) was also quite remarkable.

Many explanations have been sought for this fact, and of course the governing institutions have been examined, including the head of government, the so-called *doge* (*doxe* in the vernacular Veneto language; the term is approximately equivalent to "duke" in other languages), who was elected for life. Although initially they were chosen in a popular assembly, eventually, and during the longest period, doges were elected by an electoral college composed of 41 nobles, who were also chosen through a series of lotteries and nominations emanating from the *Maggior Consiglio*, which eventually included all noble families in Venice (Molinari 2020).

The move from the popular assembly to an electoral college Smith et al. (2021), composed of all persons with a certain economic position, or at any rate chosen by the doge, provoked many changes in the social dynamics as well as the institutional design, possibly including—according to Smith et al. (2021), who quote Maranini as mentioned in Coggins and Perali (1998) as the origin of the idea—*informal* term limits to avoid doges accumulating power individually and for their families during long tenures. Following up this law, the last years of the XII century saw the approval of other laws imposing term limits on *other* offices, from council positions to lower-ranking clerical offices Lane (1973). Moreover, at the beginning of the XIV century, a *closure* (*Serrata*) took place Puga and Trefler (2014); Rösch (2000): seats in the Maggior Consiglio were closed to only a few families, hereafter called *vecchie* or "old" families. Whether this constituted an actual closing or reduction in the amount of people eligible for office or an enlargement is still under dispute Rösch (2000); some authors Lane (1963) claim that it was simply a codification of what was already happening *de facto*. However, by introducing a legal/technical foundation for this eligibility,

it certainly contributed to the definition of a noble or patrician class, which was by many authors considered the constitutional event of the Venetian republic (Rösch 2000).

This had a series of consequences, including the creation of a (relatively) egalitarian society Stockwell (2011), at least among the nobles; according to other authors Puga and Trefler (2014), this had the opposite effect of creating a chasm between the aristocrats and the *popolani*, or common folk, which ultimately brought about the demise of the Republic. The *Serrata*, even if identified as an individual event in a point in time, can be rather considered an extended period of legal and social change that extended along more than thirty years (Rösch 2000). Rösch (2000) identifies several specific laws approved during those dates that limit election to the Maggior Consiglio to those whose fathers and grandfathers had already participated in it, and later forcing the doge to seek approval from the Council of Forty to include new members in the council. Additionally, it apparently made the extended family or *casata* a political subject by making family ties the main, if not the only, way of sharing the power of the republic (Bellavitis 2013).

At any rate, it brought stability and certainty in the succession of the head of the state, which happened invariably in a few weeks after the deceased doge was interred; it is also a fact that except for a single case, no doge had to be indicted of corruption or treason, or for that matter simply removed or murdered, as was the case in many of the other Italian city-states or Europe at large.

Although it can be claimed that the *Serrata* was not the last watershed event; some authors (Chojnacki (2000); Rösch (2000)) talk about a second and even a third *Serrata*, extending the concept to include legal changes that made the definition of nobility and other social castes evolve again (Chojnacki 2000); nevertheless, it is interesting to check what kind of intended or unintended side effects it brought into the institutional design of the Republic.

Since the way election was performed, as well as the pool of eligible candidates, cannot by themselves explain the stability and duration of the republic, we need to investigate if another factor might be at play here. Since doges were elected for life, it was apparently a conscious design by the electoral college to choose them in such a way that their terms would not last too long Smith et al. (2021). That is why, in this paper, using data and code from the `dogesr` (Merelo-Guervós 2022) R library (published in the CRAN repository, and available under a free license), we will check to what extent that happened and if there was some evolution during the time the Republic of Venice existed. We will also look at the change points in the duration of the elected rulers and try to relate them to specific events in the history of Venice.

The rest of the paper is organized as follows. Next, we will show the state of the art in studies of how doge election and related choices affected the stability of the Venetian republic. Then, we will analyze these terms and their evolution with time and check if there were some shifts in the tenure duration in Section 3. Finally, we will present our conclusions and discuss (Section 4) its implications.

## 2. State of the Art

The main reference in this area is the paper by Smith et al. (2021), which focuses on how electing doges with a certain age imposed term limits on their tenure, and also how these limits changed during war time, electing in these cases younger doges who might be more open to taking aggressive measures to protect the republic. This would make Venice an *informal* gerontocracy (Magni-Berton and Panel (2021)), a type of government generally favored for several reasons, one of which is the (wrong) perception that people whose age is above their constituency are better at making decisions that concern them.

That might be the superficial reason why Venice chose to go that way, but Smith et al. (2021) argue that by not having a term limit imposed by law, it is impossible for any ruler to change it; the time a ruler stays in power will be enforced by nature itself, as long as the pool of candidates is reduced to persons that have a certain age and health state.

Electing only doges when they have reached a certain age also has the side effect of including only candidates with a proven track record and a long life of service to the republic, which might have included (term-limited) appointments at the Senate and other institutions with executive power, such as the council of Ten. But, this does not seem to be the *reason* why they were chosen; it was simply an effect caused by the fact that doges, all of them belonging (since the Serrata) to noble families, spent their adult life in different (and term-limited) political appointments. In general, term limits pursue avoiding accumulation of power (and associated wealth) by elected officials and their extended families; informal term limits, as is the case, did have the same intended effect. In some cases, the term imposed by law was extremely short. For instance, the republic of Ragusa (currently Dubrovnik, a state that once belonged to Venice and that was heavily influenced by Venetian politics) limited terms of *rectors*—the highest office in the country, equivalent to the president of the republic—to a single month (Kunčević 2018), and they could not be reelected until two years after they had left office. Extremely limited terms also produced rotation and sharing of power between the different families that were a part of the nobility, which in the case of Ragusa were just a few dozens, which again contributed to long-term stability.

How and why this was introduced in the political mindset is subject to discussion. Smith et al. (2021) mention that the average number of years before and after 1172 changed dramatically. That was the point in time when new laws made the election the responsibility of an electoral committee, which underwent a series of changes until, after the Serrata that reduced the eligible pool to a closed set of noble families, it became a permanent fixture. These new laws also made terms for certain offices limited and non-recurrent until after a number of years. In general, laws published in the few years following 1172 brought about a complete overhaul of the political system. For Smith et al. (2021), it produced as a side effect the effective reduction of doges' rule, producing doges that were increasingly older at the moment of taking office. Their results indicate that between 1172 and 1797 (end of the Republic), the average term duration was 7.6 years, while before that it was almost double at 13.2 years. They also claim that periods of war and conflicts relaxed that informal rule, since it was considered that younger people were more daring and capable of defending the Republic or defeating the enemies; this was despite the fact that the doge that made the biggest gains for the republic, Enrico Dandolo Madden (1995), was chosen when he was (around) 85, after which he led Venice and the rest of the crusades to invade Zara and then Constantinople. His reign still lasted for thirteen years. Additionally, they mention that, after 1423, the average age when the doge was elected was raised to 70.55 and the average tenure dropped to 6.6 years.

While those claims check out with our dataset (Merelo-Guervós 2022), which was developed independently and is available with a free license, we were interested in the specific point in time when the shift in the tenure specifically occurred, and whether it was linked to the legal changes mentioned by them, or to some other legal, political or social event. In a first exploration of this hypothesis using our data (Merelo-Guervós 2022), we concluded that it was none of the specific dates mentioned in (Smith et al. 2021)—that is, neither 1172 or 1423—and that it was closer to the aforementioned Serrata.

The question we are trying to answer in this paper is if we can prove in a statistically significant way whether there was a date when the average in years of dogeship changed. Additionally, we would like to prove if there were other dates where minor shifts occurred. In every case, we will try to match those dates to events or changes in the republic. We will perform that next.

## 3. Examining Doges' Terms

Let us first examine the claims about informal limits to the doges' terms using independently collected data, the one provided by the `dogesr` package and collected independently from the one mentioned by Smith et al. (2021). How these data were collected is explained in Merelo-Guervós (2022), which does not include the exact date of election and death, just

the year; so, it is in fact impossible to know the precise number of months they were in office, only the year it started and ended. It does not include age when elected either, since that datum is irrelevant to the main focus of this paper. Thus, a term of 0 years simply indicates that both events (election and death) occurred in the same calendar year. Taking this into account, there were quite a few doges whose tenure was extraordinarily short, staying in office less than two natural years (see Table 1).

**Table 1.** Doges whose tenure was within a calendar year (indicated with 0 in the last column) or went from one year to the next. Since, due to the imprecision in the data, it is impossible to know the exact amount of months, they are grouped together in a single table.

|  | Doge | Years |
|---|---|---|
| 2 | Domenico Leone | 0 |
| 3 | Felice Cornicula | 0 |
| 4 | Teodato | 0 |
| 5 | Gioviano | 0 |
| 6 | Giovanni Fabriciaco | 0 |
| 19 | Pietro I Candiano | 0 |
| 70 | Michele Morosini | 0 |
| 106 | Nicolo Donato | 0 |
| 114 | Francesco Corner | 0 |
| 8 | Galla Gaulo | 1 |
| 28 | Vitale Candiano | 1 |
| 57 | Marino Zorzi | 1 |
| 62 | Marino Faliero | 1 |
| 64 | Giovanni Gradenigo | 1 |
| 79 | Nicolo Marcello | 1 |
| 84 | Marco Barbarigo | 1 |
| 92 | Marcantonio Trivisan | 1 |
| 99 | Sebastiano Venier | 1 |
| 108 | Francesco Contarini | 1 |
| 110 | Nicolo Contarini | 1 |
| 113 | Carlo Contarini | 1 |
| 116 | Giovanni Pesaro | 1 |
| 130 | Marco Foscarini | 1 |

All in all, 18.85 percent of all doges were in office for at most two calendar years, a remarkable number. The first column indicates a high number of them ruled at the beginning of the republic (when their cause of death was probably murder) and after the Republic changed the system of election (when, as is the main hypothesis, they were actually elected at an old age).

So, it effectively looks like many doges effectively had a short shelf life. As a matter of fact, the doge that stayed in power the longest, shown in Table 2, was a strange event that the Republic tried to ensure did not happen again.

**Table 2.** Doge that spent the most years in office and the number of years it lasted.

| Doge | Years |
|---|---|
| Francesco Foscari | 34 |

Francesco Foscari was elected when he was 50, defeating the other candidate, Pietro Loredan. After being in power for 34 years, he was the only doge forced to abdicate after the *Serrata* (new rules electing the doges explained in the introduction) and given a pension for life (Wiel 1891). In fact, he died a few days after abdication, although the stress caused by his dismissal might have contributed to his death—that is, if left unchecked, he could have been doge for a few years still.

Looking at the other end of the ranking, the 10 doges that stayed in power the longest are shown in Table 3. The first thing we may observe is that, among the top 10, there is only

one, Leonardo Loredan, who was elected years after the aforementioned Francesco Foscari. According to some historians (Rendina 1984), he was chosen by the minimum needed to get elected, even as his main opponent died unexpectedly during the election process. So, he was really an outlier, although according to Smith (2020), doges with a longer *shelf life* should be entirely expected in terms of turmoil, which was the case at the turn of the XVI century; it might have happened, however, that the change point that brought informal term limits was the bad experience with Foscari. This ranking, however, shows how very long terms were extremely infrequent, and there were no doges that ruled for more than twenty years after 1521, when Leonardo Loredan died.

**Table 3.** The top ten doges ranked by tenure length, the precise years where they acceded to power and left, and the total number of years.

| Doge | Century | Start | End | Years |
|------|---------|-------|-----|-------|
| Francesco Foscari | 14 | 1423 | 1457 | 34 |
| Domenico I Contarini | 10 | 1041 | 1071 | 30 |
| Pietro Tradonico | 8 | 837 | 864 | 27 |
| Pietro Tribuno | 8 | 888 | 912 | 24 |
| Pietro Ziani | 12 | 1205 | 1229 | 24 |
| Maurizio Galbaio | 7 | 764 | 787 | 23 |
| Pietro Gradenigo | 12 | 1289 | 1311 | 22 |
| Orso II Participazio | 9 | 912 | 932 | 20 |
| Jacopo Tiepolo | 12 | 1229 | 1249 | 20 |
| Leonardo Loredan | 15 | 1501 | 1521 | 20 |

The histogram with 5-year wide bins shown in Figure 1, shows how infrequent, in fact, were terms of more than 10 years. The distribution is skewed towards very short terms because it also includes the first years of the institution, when murder or destitution by popular revolt was relatively usual; it also shows a relatively long tail exactly for the same reasons, but since it is clustered in the shorter terms, it is quite clear that the limitation (Smith et al. 2021) talk about is indeed at work.

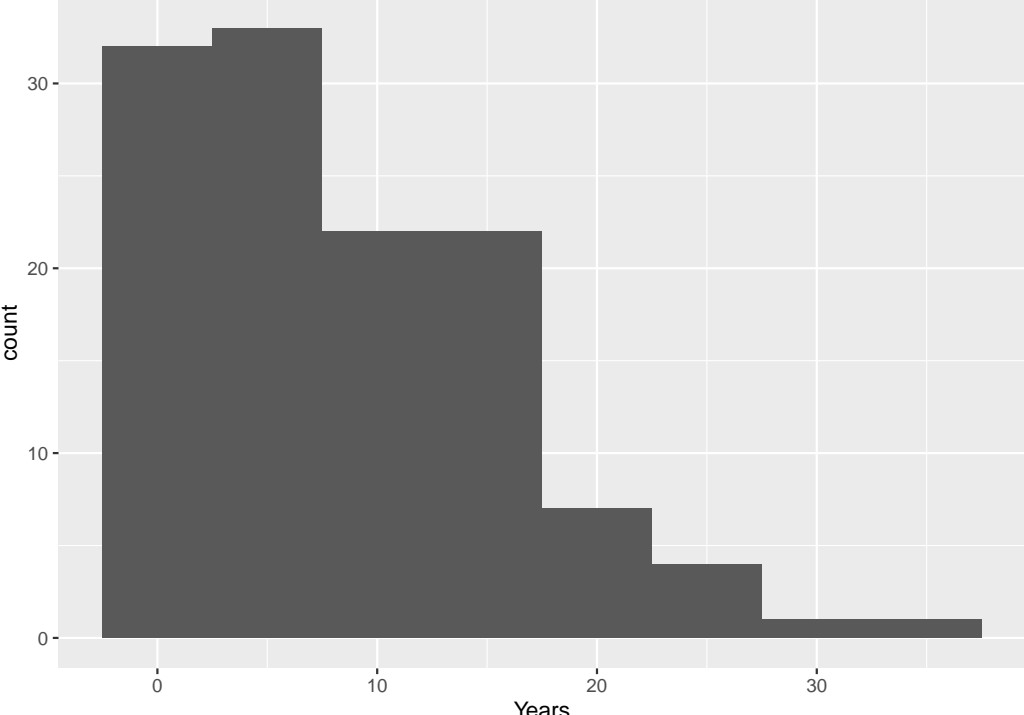

**Figure 1.** This histogram reproduces, with a fixed bin size, Figure 1 in Smith et al. (2021).

Smith et al. (2021) also indicate that an informal term limitation was evident at a certain point in time, with big differences before and after 1172. There is no claim, however, that the *cause* of that term limitation occurred in that moment, or that dismiss any other possible date or event as the trigger. We will try, in this paper, to check if it is at all possible to find the point of change and match it to some specific or significant event. Let us start checking that fact by looking at the average duration in office through the history of the Republic, shown in Figure 2.

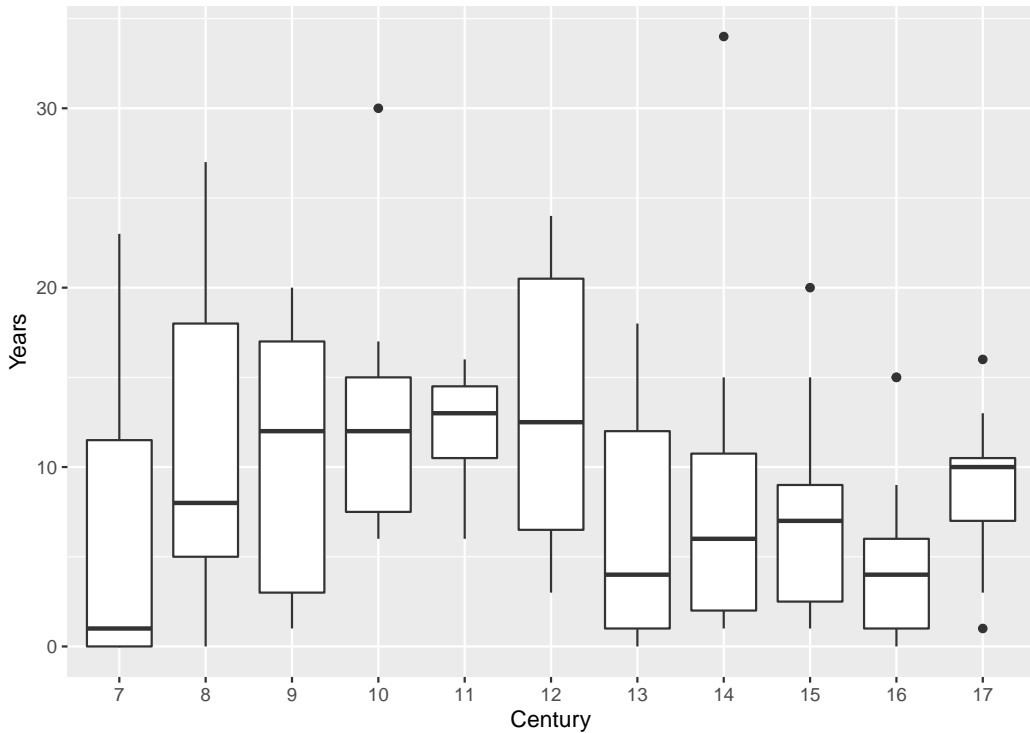

**Figure 2.** Boxplot of doge duration in office vs. century. As is the convention, black dots represent outliers.

There seems to be a clear difference before and after the fourteenth century (indicated by the two first digits of the century, 13, in the chart); there is also a return to longer tenures by the end of the Republic, in the 18th century; in fact, centuries X to XIII show a remarkably stable average of around 12 years from election to death.

This chart also shows that, apparently, the change from the XII to the XIII was, on average, small; the change was rather on the high and low quartiles, with a higher variation in the amount of time a doge lived[1]. After observing this apparent shift, we would like to investigate a bit more thoroughly when that change happened. In our previous report (Merelo-Guervós 2022), we showed that the difference between the average term pre- and post-Serrata is *higher* than the one indicated by Smith et al. in their paper; effectively, Serrata laws were introduced in the early XIV century, when the shift in Figure 2 is observed.

In this paper, however, we will analyze, using statistical methods, when the actual change points in the time series that include all the terms, and which we have already summarized above, occurs. Change-point detection algorithms (Truong et al. 2020) indicate the precise moment when a structural change occurs, so that differences in averages before and after the change are maximized; in this case, we will be measuring when the average doge term changes. There are many such algorithms that use different statistical tests.

Using Lanzante's test (Lanzante 1996) from the `trend` R package (Pohlert 2020), this change occurs in the moment Francesco Dandolo took office. Francisco Dandolo ruled from 1329 to 1339, right after the last laws of the Serrata were enacted[2]. Other tests also yield the

same change point. See Figure 3 and Table 4 for a boxplot of the difference, in years, of the doges' term pre- and post-Serrata.

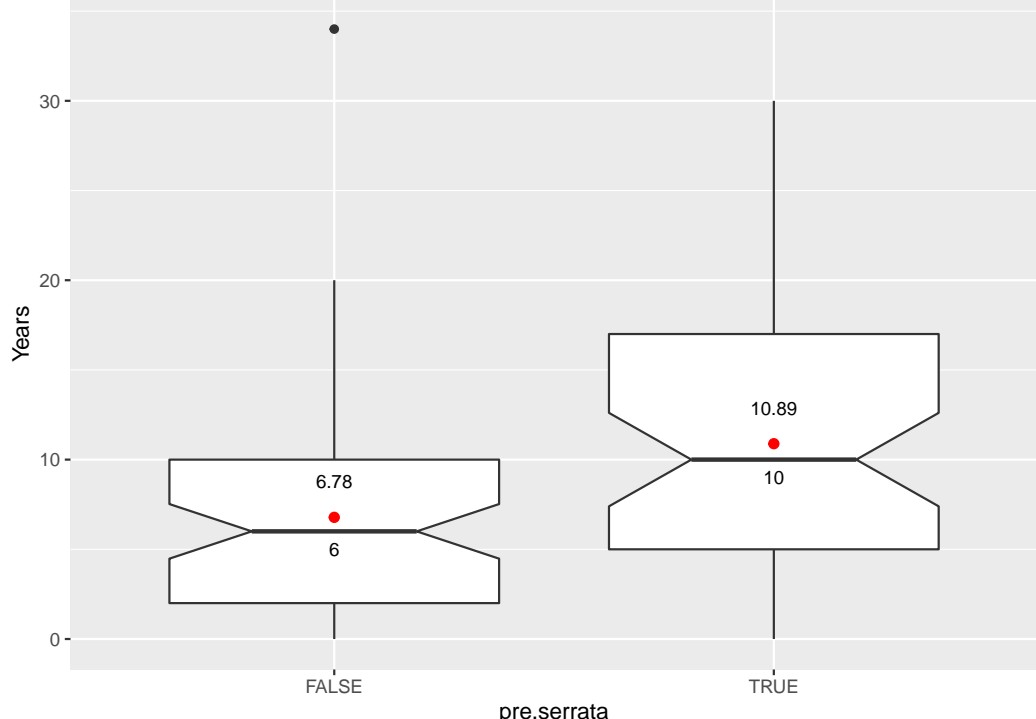

**Figure 3.** Box-and-whisker plots of the span, in years, of doges in office pre and post 1329 (indicated with the pre.serrata variable). The red point and legend indicates the average, and the median is also shown in the chart; as is usual, black dots represent outliers.

**Table 4.** Median and average span, in years, of the doge rule, pre and post the found change point.

| Period | Average | Median |
|--------|---------|--------|
| Post-1329 | 6.782609 | 6 |
| Pre-1329 | 10.886793 | 10 |

There is a difference of more than four years in the average, and median, pre-and post Serrata, although this effect was not immediate and took a few elections to sink in; in fact, as indicated in the introduction, the Serrata included a series of laws passed during a 30-year period; the end of this 30-year period is very close to the change point found here. Please note that both quantities are lower than the ones published in Smith et al. (2021): 13.2 before 1172, and 7.6 after, with a difference of 5.6, as opposed to a difference in averages of 4.1 before and after the Serrata.

This, in fact, points to other possible change points present in the time series. Let us try and analyze, separately, these two periods, looking for other change points. First, we examine the pre-Serrata period. We would be interested in checking if, after all, the introduction of the electoral college effectively had some measurable and significant effect.

The Pettitt test (Pettitt 1979), which works better for smaller series, yields a change point at index 9 (after Galla Gaulo, who was blinded and exiled after a year in power); however, this result has a p-value higher than 0.05 and, thus, we need to conclude that there is indeed no significant structural change in this period, not even after the introduction of the electoral college, as singled out by Smith et al. (2021).[3] As a matter of fact, looking at Figure 2, we see that the VIII century had exceptionally short terms; it increased slightly during the IX century, but it then stabilized until the change point in the XIV century, when it showed a clear decrease, which is statistically significant. Other tests, the Buishand and

Buishand U test, stop at index = 28; however, again, the p-value is relatively high. Doge number 28 in our dataset would also happen before the year 1000, and thus, again, does not match the date indicated by Smith et al. (2021).

However, there seems to be some (possibly) significant change by the end of the republic, at least according to Figure 2, which shows a shift from a median that is lower than 5 years, to a median equal to 10 during the XVIII century. Let us look at the series that starts with Francesco Dandolo while trying to identify if there are actually some statistically significant change points.

In this case, we use another change-point detection test which is more adequate in this case—Pettitt's test (Pettitt 1979). This indicates that the change point occurs with the Giovanni Pesaro tenure, from 1658 to 1659 (Rendina 1984). This was an election quite unlike others: Giovanni Pesaro had been a corrupt administrator, and married a non-patrician to boot. However, his hawkish attitude and the fact that he pledged his own money to support the war against the Ottoman empire tipped the balance in his favor, making him possibly the first doge to be elected not because of his qualities (and age) but because of his money. The republic at the time of him taking office, 1658, was already in terminal decline, and although he ruled for a single year, possibly precipitated this change in regime.

The statistical distribution of terms before and after this presidency, plotted in Figure 4, shows a smaller difference in medians of 2.5 years; the difference in averages, with respect to the one shown in the main change point, also decreases. There is a clear overlap between the two statistical distributions, and the Pettitt test has a p-value that is relatively high. As a matter of fact, a non-parametric Wilcox test shows no significant difference. We could then dismiss this change point as not significant, although it can explain the greater duration of the doges' terms elected by the last period of the republic. If we compare the period between the Serrata and this candidate to the change point and the pre-Serrata era, the Wilcox test yields a p-value of 0.004499, which is quite low indeed, indicating a *very* significant difference, implying that excluding this last period of the republic when the informal term limits apparently disappeared makes the difference in terms of pre- and post-Serrata even more significant.

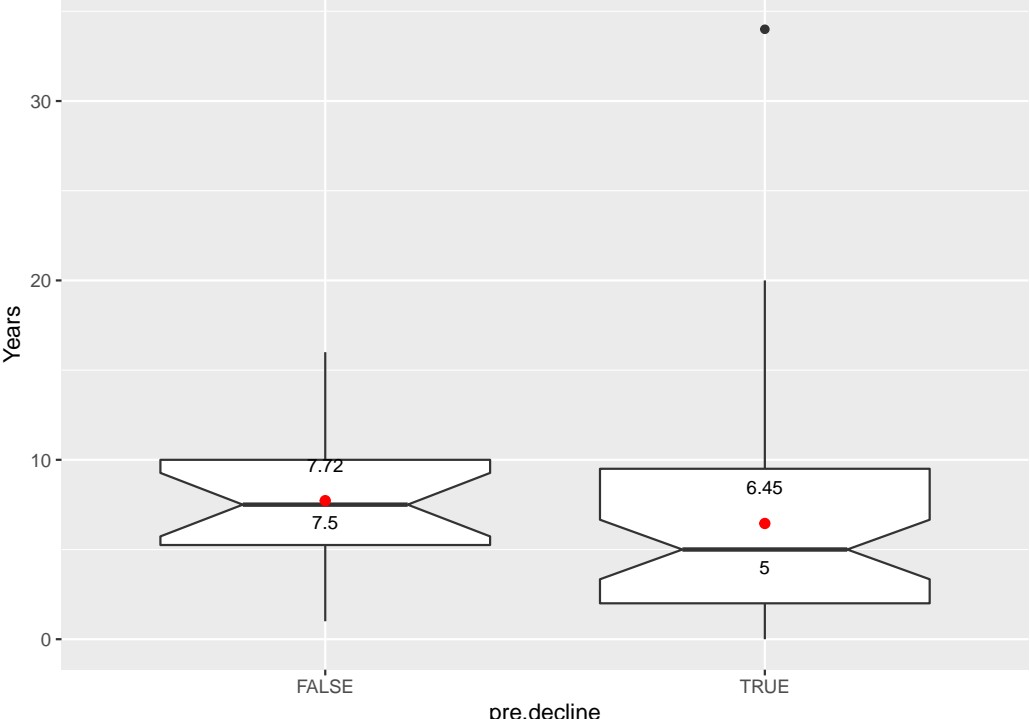

**Figure 4.** Box-and-whisker plots of the span, in years, of doges in office after 1329 and pre and post 1658 (indicated with the pre.decline variable); black dots represent outliers.

## 4. Conclusions and Discussion

As indicated by Smith et al. (2021), there seems to be evidence that term limits of doges of the Republic of Venice were actually enforced, albeit not in a formal way. In their paper, they show there was a significant difference between terms before and after 1172, when the Maggior Consiglio was formed, which implied a shift from popular vote to a vote by an electoral college. The statistical analysis that we have performed in this paper effectively indicates that, at a certain point in time during the history of the Republic of Venice, there was a conscious choice by the electoral college to choose, among the eligible candidates, those that were beyond a certain age, and thus with a limited expected lifespan.

Using a statistical technique called change-point analysis, which computes the point in a series where the average changes in a statistically significant way, several algorithms support the moment in which the election of Francesco Dandolo took place as the moment where the shift in the central tendencies, the median and the average, took place; no significant structural shift occurred either before or after that, although there are smaller, non-significant ones that support a certain segmentation of this "tradition". So, we could say that between 1328 and 1658 (when Pesaro was elected), most nobles sitting in the final stages of the electoral college were aware of it, and elected older persons to office, which resulted in doges with a median remaining life span of just 5 years.

How this informal measure in this reduced time period contributed to the stability of the state at large is difficult to say. However, we can discuss that the fact that it happened precisely after the Serrata points to a rationale among electors that is complementary to the ones claimed by Smith et al. (2021)—that is, avoiding entrenchment in power and corruption due to long terms in office.

What happened in the Serrata is that the pool of eligible families was reduced to just two dozen. This pool was continuously expanded; still, at any point in time, there were just a few males in every family who could even aspire to be candidates due to age, offices already held, or other qualifications such as military experience. If doges would have been chosen for an extended and possibly unpredictable term due to their young age, the probability of becoming doge in every generation was quite diminished. On the other hand, by having such a high churn in the highest office, any candidate who was defeated in an election was relatively likely to be chosen in the *next* one taking place a few years (less than six, on average) down the line. This simple fact probably kept powerful families from staging a coup, which was riskier than simply waiting for nature to follow its course. "What goes around, comes around", they could think. With a very small set of families, their turn would eventually come sooner or later, probably and possibly in the same generation.

The single fact of changing from election in a popular assembly to instituting an electoral college did not really have that effect. Electing a person who could stay in power for many years did not change the possibility of having a person in your own family elected in a significant way. Since the pool was larger (any family of a certain standing in the city, a pool that changed continuously), the possibilities stayed more or less the same and close to zero. Someone in your family could or could not be elected next, but since there were *many* other possible candidates, it was really indifferent that this happened many years later, since it would not be the *turn* of your family no matter what.

The number of families inscribed in the Golden Book increased with time, however. By the XVII century, anyone could pay their way into the Maggior Consiglio, thus increasing the number of possible families to choose from and decreasing the incentive of choosing doges for short terms. How this influenced the length of the terms might be a future and interesting line of work that might be pursued in future papers.

A more sensitive analysis could also be pursued in the future. In their paper, Smith et al. (2021) examine also how the presence or absence of conflict could have an influence in the age of the elected doge; that could be one additional line of research but, along the lines of the Merelo-Guervós (2022) report, the position of the candidate in the social network could also play an additional role in this choice.

It would be interesting to check whether other republics with similar systems, such as the Vatican, the republic of Ragusa or the republic of Genoa, have a similar informal term limit and what kind of factors influenced it. Further research, as well as data gathering, will be needed to prove or disprove these hypotheses and show similarities (or disparities) with the Venetian system.

**Funding:** This paper has been supported in part by project and DemocratAI PID2020-115570GB-C22.

**Institutional Review Board Statement:** Not applicable

**Informed Consent Statement:** Not applicable

**Data Availability Statement:** This paper has been written using the literary programming system `knitr`, and all analyses performed are embedded in its source code, which is available at GitHub https://github.com/JJ/dogesr/blob/main/reports/doges-terms.Rnw (accessed 11 January 2023) under a free license. As indicated, it uses as the data source version 0.1.1 of the CRAN `dogesr` package and it can be compiled using RStudio or `pandoc` after installing the needed packages.

**Conflicts of Interest:** The authors declare no conflict of interest.

## Notes

[1] This can be explained within Smith et al. (2021) framework, who stated that in times of conflict, younger doges were chosen. Certainly, the XIII century was a time when the Republic was almost continuously at war; notably, the IV Crusade that ended with Venice taking over Constantinople: short periods of peace with "old" doges, followed by long periods of conflict with "younger" doges, such as Enrico Dandolo Madden (2006). However, this effect should not stop by the end of the century and would affect the rest of the elections; this is why we are interested in checking whether there was an change in the average, including the exceptions to the rule in times of conflict.

[2] Please see also refer to Merelo-Guervós (2022) for the importance of the Dandolo family in the social network of Venetian noble families.

[3] This does not mean that the laws introduced in 1172 themselves were not conducing to structural change, only that the change cannot be observed until much later, after more changes were introduced, because the period between both events was characterized by a high level of conflict, which, again according to Smith et al. (2021), implies selecting younger candidates with more *elan*. To a certain point, the observed shift in structural change confirms, rather than rectifies, the main thesis of that paper.

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
