# Peer review of "It’s a Doge’s Life: Examining Term Limits in Venetian Doges’ Life Tenure"

_2409-9252, doi:10.3390/histories3010003_

Round 1

Author Response

We are very grateful for their complete and insightful review. We will try to address your concerns (highlighted) below:

The author could readily address this by simply noting that their argument was not inconsistent with the Smith, Crowley, and Leguizamon (2021) paper (and that the data still indicates that it is operative even if only the post-1329 data are examined).

You are absolutely right, and we make no attempt to rebut the affirmations in that paper. We indicate where our results are complementary, and also where they add a possible explanation to term limits.

The evidence for this contention is on shakier ground than the author
admits.

Statistical evidence can hardly be called "shakier". However, we have now inserted an explanation that would make these results consistent with Smith et al. hypothesis, which, as we said above, don't try to overturn, but complement and analyze in a rigourous way. The period between 1172 and 1329, as you indicate, is characterized by conflict; we know say so explicitly in the paper; that might explain why the statistically measurable shift happened after the Serrata, which is generally admitted by many authors (such as the referenced Puga & Treffler) to have been a very significant event in the existence of the republic, and would effectively provide even more reasons to informally limit doges terms.

The anecdotal historical evidence also strongly favors the shift being in 1172, as Doge Vitale II Michiel, elected in 1156, was famously assassinated after a long reign, leading to major constitutional reform.

We don't argue against this in our new version of the paper. The Serrata could not have happened without a shift from "popular" vote to an electoral college. However, there are many factors that contribute to a shift, and they possibly accumulated at the time of the Serrata (understood as an extended period that saw the publication of many different laws related to eligibility to official posts and dogeship, as now indicated in the introduction). However, as you say, there were only 8 doges and one of them was Enrico Dandolo, which leads to a very high variation in terms (as evidenced in the very high standard deviation shown in Figure 2) that made the actual shift, measured statistically, to take place later.

This would also point out to indirect, or maybe long-term, effects: electoral college lead to less conflicts, but also a differentiation between different strata of wealthy families, which eventually led to the Serrata and actual evidence of term limits due to the accumulated effect of less conflicts (thus actual enforcement of the informal rule that elected older doges) and, as we point out in the paper, a reduced pool of candidates that would enhance the probability of a specific candidate being elected if defeated in an election.

I think the author could moderate their claim to be that the evidence they present suggests that the electoral norm of selecting elderly doges was strengthen and enhanced with the Serrata in 1329

We have done so, and we are grateful for the suggestion

Reviewer 2 Report

This paper follows Smith et al (2021) in investigating tenure length for doges in historic Venice. The author applies statistical techniques to attempt to identify the point at which doge tenure underwent a structural break. The author identifies possible points later than 1172, which was the earliest point considered in the Smith et al (2021) paper.

This paper is interesting in that it attempts to shed additional light on the questions raised originally in Smith et al (2021). The biggest issue with the analysis, however, is the contention that the Smith et al (2021) paper identifies 1172 specifically as some pivotal point. A careful reading of that paper shows that while 1172 is the earliest period in the data considered, it is not clear that the authors were arguing this year represented a pivotal change in Venetian doge tenure. In fact, the main argument of that paper is that ducal tenure fluctuated depending on other factors occurring during the time period considered.

Thus, while the paper's analysis appears competently executed, and its findings (especially the suggested explanations offered in the conclusion) are interesting, I believe the author could better frame this paper as an extension of the Smith et al (2021) piece rather than an attempt to "prove" or "disprove" any aspect of their thesis. As it reads now, the author claims to have disproven their thesis, which is a misstatement of that earlier paper's core argument.

The true contribution of this paper is its taking a more detailed and nuanced look at changes in ducal tenure during the time period in question, and raising interesting potential areas for future research.

Minor comments:

There are several instances where nonstandard English or other typographical errors make the paper difficult to understand. For instance, on page 3, "In fact he died a few days later, although the stress caused by his demise might have contributed to that." A thorough proofreading should be completed.

The author cites an "article" from Wikipedia, alongside a claim, "...according to everyone." Wikipedia is not generally considered a valid source for scholarly work.

Table 1 is formatted awkwardly such that the bottom row appears disjointed with the rest of the figure.

Author Response

I'll try to address your points, which I insert here highlighted.

This paper is interesting in that it attempts to shed additional light on the questions raised originally in Smith et al (2021). The biggest issue with the analysis, however, is the contention that the Smith et al (2021) paper identifies 1172 specifically as some pivotal point. A careful reading of that paper shows that while 1172 is the earliest period in the data considered, it is not clear that the authors were arguing this year represented a pivotal change in Venetian doge tenure. In fact, the main argument of that paper is that ducal tenure fluctuated depending on other factors occurring during the time period considered.

That is absolutely correct. Even if they use that date as base for computing averages in age, and identify it as a turning point in the history of the Republic, they don't claim to be the point, and as such, our paper is rather complementary, not a rebuttal of those claims.

The introduction and state of the art have been largely rewritten to reflect more accurately this fact, as well as introducing information on the changes that occurred in the dates mentioned by Smith et al.

Thus, while the paper's analysis appears competently executed, and its findings (especially the suggested explanations offered in the conclusion) are interesting, I believe the author could better frame this paper as an extension of the Smith et al (2021) piece rather than an attempt to "prove" or "disprove" any aspect of their thesis. As it reads now, the author claims to have disproven their thesis, which is a misstatement of that earlier paper's core argument.

As indicated above, this has been largely corrected; the main point of the paper, the change in doges' election process which made the electoral college choose elderly patricians happening mainly after the "Serrata" now stands out more clearly, and we appreciate this review for this insight.

[...]

Minor comments:

There are several instances where nonstandard English or other typographical errors make the paper difficult to understand. For instance, on page 3, "In fact he died a few days later, although the stress caused by his demise might have contributed to that." A thorough proofreading should be completed.

Corrected what was mentioned, and done an extensive proofreading. Thanks.

The author cites an "article" from Wikipedia, alongside a claim, "...according to everyone." Wikipedia is not generally considered a valid source for scholarly work.

That is so true. I have followed the references in the Wikipedia article, and assigned claims to the precise document. More historical references have been also added, mainly responding to another reviewer, but in general the introduction and state of the art now reflect a more comprehensive panorama of historical research in the Republic of Venice and related to term limits in political appointments.

Table 1 is formatted awkwardly such that the bottom row appears disjointed with the rest of the figure.

All tables have been reformatted and captioned. As a matter of fact, what seemed to be a table row was in fact a totally different table; now it shows.

Reviewer 3 Report

The premise of this article and its methodology is novel and interesting. The article, however, would materially benefit with engagement with the robust scholarly literature on the Venitian Republic which hasn't been cited in this article. The reviewer recommends the following:

Venice Reconsidred: The History and Civilization of an Italian City-State

A Companion to Venetian History 1400-1797

Without engagement with this literature, it is hard to evaluate the strength of this study. The reviewer recommends that the authors engage with this literature then evaluate their findings and methodology. 

Author Response

We are very grateful to this reviewer, for the helpful suggestion. This is our first paper in a history journal, so we really appreciate some methodological insight that can help our research, and obviously publication of the paper.

We have effectively checked the references mentioned, especially the chapters that were more related to the events that are mentioned in the paper:

  • Bellavitis, Anna. 2013. A companion to Venetian History, 1400-1797, Eric Dursteler, ed., Chapter Family and
    Society, pp. 342–365. Brill.
  • Chojnacki, Stanley. 2000. Venice Reconsidered: history and civilization of an Italian city-state, 1297-1797; John Martin and Dennis Romano, editors, Chapter Identity and ideology in Renaissance Venice: The third
    serrata, pp. 263–94. Johns Hopkins University Press.
  • Rösch, Gerhard. 2000. Venice Reconsidered: history and civilization of an Italian city-state, 1297-1797; John Martin and Dennis Romano, editors, Chapter The Serrata of the Great Council and Venetian Society, 1286-1323, pp. 67–88. Johns Hopkins University Press.

Additionally, following up references from these, an additional source has been identified:

  • Lane, Frederic C. 1963. Recent studies on the economic history of Venice. The Journal of Economic History 23(3), 312–334. https://doi.org/10.1017/S0022050700104097.

The introduction has been expanded with a more thorough explanation of what the Serrata meant for the social, political (and also mercantile) dynamics of the republic; I think this further supports the proved fact that there was a change in the age at which doges were elected caused by the set of legal changes that have collectively called the Serrata (mainly by Rösch)

Round 2

Reviewer 2 Report

I thank the author for his careful consideration of my previous comments. I believe the paper is much improved.

My only remaining comment is that some of the additions have added additional typographical errors to the manuscript. I would recommend an additional proofreading.

Reviewer 3 Report

The revisions to the article greatly improved the argument and structure.